# Unveiling the Anatomical and Functional Attributes of Stipular Colleters in *Palicourea tetraphylla* Cham. & Schltdl. and *Palicourea rudgeoides* (Müll. Arg.) Standl. (Rubiaceae)

**DOI:** 10.3390/plants13091206

**Published:** 2024-04-26

**Authors:** Laís de Almeida Bezerra, Emilio Castro Miguel, Camilla Ribeiro Alexandrino, Thaiz Batista de Azevedo Rangel Miguel, Valdirene Moreira Gomes, Maura Da Cunha

**Affiliations:** 1Center for Biosciences and Biotechnology, Laboratory of Cellular and Tissue Biology, Universidade Estadual do Norte Fluminense Darcy Ribeiro, Campos dos Goytacazes 28013-602, RJ, Brazil; laisbezerraa@outlook.com (L.d.A.B.); camillalexandrino@yahoo.com.br (C.R.A.); 2Department of Metallurgical and Materials Engineering, Biomaterials Laboratory, Universidade Federal do Ceará, Fortaleza 60440-900, CE, Brazil; emilio.decastromiguel@gmail.com (E.C.M.); thaizrangel@gmail.com (T.B.d.A.R.M.); 3Center for Biosciences and Biotechnology, Laboratory of Microorganism Physiology and Biochemistry, Universidade Estadual do Norte Fluminense Darcy Ribeiro, Campos dos Goytacazes 28013-602, RJ, Brazil; valmg@uenf.br

**Keywords:** plant anatomy, secretory structure, secretory process, microorganisms, ultrastructure, secretion, programmed cell death, morphoanatomy

## Abstract

The characterization of colleters in Rubiaceae is crucial for understanding their role in plant function. Analyzing colleters in *Palicourea tetraphylla* and *Palicourea rudgeoides* aims to deepen the understanding of these structures morphoanatomical and functional characteristics. The study reveals colleters with palisade epidermis and a parenchymatic central axis, classified as standard type, featuring vascularization and crystals. Colleter secretion, abundant in acidic mucopolysaccharides, proteins, and phenolic compounds, protects against desiccation. The ontogenesis, development, and senescence of the colleters are quite rapid and fulfill their role well in biotic and abiotic protection because these structures are present at different stages of development in the same stipule. Pronounced protrusions on the colleters surface, coupled with the accumulation of secretion in the intercellular and subcuticular spaces, suggest that the secretory process occurs through the wall, driven by pressure resulting from the accumulation of secretion. The microorganisms in the colleters’ secretion, especially in microbiota-rich environments such as the Atlantic Forest, provide valuable information about plant–microorganism interactions, such as resistance to other pathogens and organisms and ecological balance. This enhanced understanding of colleters contributes to the role of these structures in the plant and enriches knowledge about biological interactions within specific ecosystems and the family taxonomy.

## 1. Introduction

Secretory structures emerge as multifunctional elements in plant life, imparting remarkable plasticity and adaptability of plants in various environmental conditions. Their diverse functions are directly related to morphological variety, distinct anatomical locations, the period of activity, and the specific chemical composition of the released exudate [1,2]. In this context, colleters are highlighted as structures with significant potential for revealing fundamental plant biology and evolution aspects.

The term colleter refers to the sticky substances secreted by these secretory structures [3]. Their secretions exhibit notable chemical complexity [4,5,6,7,8,9,10,11,12]. Additionally, the composition of these secretions varies at different stages of development, adding further complexity to the phenomenon [13,14]. This secretory structure has the function of protecting the leaf primordium and meristem against biotic and abiotic damage [15]. Studies have highlighted the connection between environmental parameters and the composition of colleters’ secretion [16,17].

Colleters are present in over 60 angiosperm families [3]. Within the Rubiaceae family, these colleters are often associated with the adaxial surface of interpetiolar stipules, standing out as diagnostic elements of this botanical family [18,19]. The Rubiaceae family exhibits morphological complexities that contribute to taxonomic challenges, including discrepancies in the delimitation of subfamilies, tribes, and genera. The morphological diversity and the adaptation of Rubiaceae specimens to various environments result in many taxa, making their delimitation and separation challenging, as some may show intermediate characteristics.

The evolution of research on the anatomical, histochemical, senescence, and ultrastructural characterization of colleters in Rubiaceae has provided valuable data, establishing itself as a fundamental tool for a more comprehensive understanding of these structures [6,7,12,20,21,22,23,24].

A previous study assigned two distinct roles to the colleters of a species within the genus *Palicourea* and their secretions: antifungal action and the ability to attract and maintain a conducive environment for the proliferation of beneficial organisms [15]. It is crucial to emphasize that the literature presents a wide diversity of alkaloids in the genus *Palicourea*, a key aspect for understanding the ecological significance and evolution of the secondary metabolism within the group [25]. 

This study aimed to address knowledge gaps within the Rubiaceae family, focusing on two endemic species, *Palicourea rudgeoides* and *Palicourea tetraphylla*, found in the southeast of Brazil, in the Atlantic Forest [26]. This study describes the developmental stages of colleters, from inception to senescence, and delves into the characteristics of the exudates they produce. Additionally, the research examines the functional role of colleters, considering their structure and exudate content within these plants’ ecological context.

## 2. Results

For the identification of stipular colleters in the species under analysis, the following terminology was adopted: stipules at the shoot apex were designated as first node stipules, followed by the subsequent stipules of the second node, and so forth. In the case of *P. tetraphylla*, stipules were collected from the first to the third node, while in *P. rudgeoides*, the collection encompassed from the first to the fifth node.

At the shoot apex of *P. tetraphylla*, a covering of white secretion was evident (Figure 1A), with a notable quantity of clustered colleters ordered in a triangular arrangement (Figure 1B). Colleters from the first node displayed a white coloration (Figure 1C), indicating early stages of development. A brown coloration was observed at the colleters apex (Figure 1D), signaling the early senescent phase. In the second node, colleters were identified in the early and advanced senescence stages (Figure 1E,F). In the third node, most colleters had acquired a brown or dark coloration, although some in a young stage were still discernible (Figure 1G–I).

The secretion was hyaline and scanty at the shoot apex of *P. rudgeoides* (Figure 1J). The colleters of this species are finger-shaped and have a similar distribution in *P. tetraphylla* (Figure 1K–O). Colleters from the first node exhibited a yellowish coloration, and in some instances, a brown hue was noted only at the apex of the structure (Figure 1K,L). In the third, fourth, and fifth nodes, it was observed that senescence in the colleters also manifested a dark coloration (Figure 1M–O).

A peculiar organization was observed when examining the shoot apex of *P. tetraphylla* by scanning electron microscopy. The leaf primordia were surrounded by a pair of stipules containing trichomes and cylindrical colleters at the base of the stipule (Figure 2A). Upon closer inspection of the colleters it was evident that their surfaces were turgid (Figure 2B).

In the stipules of the second and third nodes, colleters with wrinkled apices (Figure 2C–E) and others with a turgid appearance (Figure 2F) were noted. The surface of the colleters of the second and third nodes showed protuberances (Figure 2G–I).

*P. rudgeoides* colleters showed a gradual change in their surface along subsequent nodes, unlike *P. tetraphylla*, which exhibits colleters at different stages at the same node. In a frontal view of first node colleters, the outline of the cells through the cuticle was observed (Figure 2J). From the second to the fourth node, the accumulation of secretion in the subcuticular space caused distension of the cuticle to different degrees, generating bulges on the surface (Figure 2K,L). Colleters from the fifth node also showed these bulges; however, they only showed the wrinkled apex or a significant part of its extension (Figure 2M,N). Tectors trichomes were observed around the colleters in both species.

Under light microscopy, transverse and longitudinal sections of *P. tetraphylla* colleters revealed an organization with a central parenchymatic axis surrounded by an epidermal palisade layer (Figure 3A–C). Colleters from the first node (Figure 3A) exhibited turgid epidermal cells and no accumulation of secretion, while in colleters from the second and third nodes, secretion accumulation was observed around the intercellular spaces (Figure 3B–E). This pattern of accumulation of secretion and vascularization of the central axis (Figure 3G) was observed in all nodes. In the second node, it was possible to identify young (Figure 3B,D) and senescent (Figure 3C,D) colleters, with the latter cells completely degraded. Numerous styloid crystals and raphid bundles were visualized in the central axis (Figure 3F,H).

The colleters of *P. rudgeoides* also displayed an organization with a parenchymatous central axis surrounded by a palisade epidermal layer (Figure 3I–P). Colleters from the first node exhibited turgid cells and a slight accumulation of secretion in the intercellular spaces (Figure 3I). This accumulation increased in the colleters from the second node, along with the presence of secretion in the subcuticular area (Figure 3J). All colleters from the third node were already senescent, with disorganized cells in both the axis and the epidermis and apparent cytoplasmic degradation (Figure 3L). This process continued in the colleters of subsequent nodes (Figure 3M,N). The presence of crystals in the central axis was evident, as well as the vascularization of this central axis in the colleters of this species (Figure 3O,P).

The histochemical analyses of the colleters of *P. tetraphylla* and *P. rudgeoides*, the reaction with Ruthenium Red indicated the abundant presence of mucopolysaccharides in the secretion (Figure 4A,I). Tests with the Nadi reagent (Figure 4B,J) and Sudan IV (Figure 4C,K) revealed that the colleters are coated with a cuticle. The metachromatic reaction with toluidine blue showed that the secretion contains basic compounds, evidenced by the bluish-purple coloration (Figure 4D,L). The use of Coomassie Brilliant Blue highlighted the presence of proteins in the secretion (Figure 4E,M). The test for phenolic compounds with ferric chloride was positive only in the colleters of *P. tetraphylla* (Figure 4F). The reaction with rubeanic acid was negative in young colleters (Figure 4N) and positive in senescent colleters (Figure 4G,O), indicating that the accumulation of fatty acids occurs only in this senescence phase. Applying the metachromatic toluidine blue reaction in senescent colleters revealed a change from basic to acidic character, with the color transitioning from bluish-purple to green, characterizing senescent tissues (Figure 4H).

In the ultrastructure of the secretory cells of the colleters from the first node of *P. tetraphylla*, there was an evident nucleus with a clear distinction between euchromatin and heterochromatin and the nuclear envelope with a preserved appearance (Figure 5A). The peripheral cytoplasm exhibited a large vacuole, a rough endoplasmic reticulum concentrically arranged (Figure 5B), mitochondria, and plastids (Figure 5C). However, even in the first node, the secretory cells showed signs of senescence. Moreover, some secretory cells contained microorganisms, which we identified as bacteria, inside thoroughly degraded cytoplasm (Figure 5D). In the second and third nodes, some secretory cells displayed a nucleus with irregularly condensed chromatin and a structurally altered nuclear envelope (Figure 5E), which points to cell death. Accumulation of amyloplasts was also observed (Figure 5F) and mitochondria (Figure 5G). However, most secretory cells had microorganisms inside and degraded cytoplasm, making it impossible to distinguish organelles, including the nucleus (Figure 5H,I). Due to this fact, the characterization of the programmed cell death process of the colleters of this species was not conducted.

In *P. rudgeoides*, the secretory cells of the first node also exhibited peripheral cytoplasm, a large vacuole with intact membrane in some cells (Figure 5J), rough endoplasmic reticulum, a slightly developed Golgi complex, numerous mitochondria, and a conspicuous nucleus with clear distinction between euchromatin and heterochromatin, along with a preserved nuclear envelope (Figure 5K). In some cells, the presence of plasmodesmata on anticlinal walls was noticeable (Figure 5L).

In the second and third nodes, rough endoplasmic reticulum cisternae near the nucleus and the cell wall were observed. Lipid droplets in the cytoplasm were also notable, along with numerous mitochondria clustered at the nuclear periphery and in the cytoplasm. Plastids with electron-lucent content were frequently observed. The secretion was characterized by containing numerous microorganisms (Figure 5M). In the fourth and fifth nodes, the cytoplasm of the secretory cells was already degraded, with indistinguishable organelles (Figure 5N). The nucleus appeared to be deformed when observed with the nuclear envelope rupturing (Figure 5N). Microorganisms were also observed in the cytoplasm of these nodes (Figure 5O).

In *P. rudgeoides*, it was possible to observe a gradual change in the ultrastructural characteristics of the colleters in subsequent nodes and witness their senescence process. The positive TUNEL test for the colleters secretory cells of the first node showed that they already had highly fragmented DNA (Figure 6A–C). In contrast, the negative TUNEL tests for the colleters from the second node onwards indicated that the DNA was already completely degraded in these nodes (Figure 6D–F).

## 3. Discussion

As commonly observed in most Rubiaceae species, *P. rudgeoides* and *P. tetraphylla* display colleters on the adaxial stipules surface of their stipules [18,20,21]. In the studied species, the colleters are arranged in a triangular pattern at the stipules base, a distribution documented within the family [6,14]. However, this characteristic varies across species; for example, in *Simira pikia* K. Schum., *Psychotria leiocarpa* Cham. & Schltdl., and *Psychotria suterella* Müll. Arg., the colleters are arranged in a line at the base of the stipule [14]. *Simira rubra* Mart. exhibits colleters distributed in multiple lines, while *Psychotria carthaginensis* Jacq. forms a vertical line. In *Psychotria ruelliifolia* Cham. & Schltdl. and *Psychotria capitata* Ruiz & Pav., the colleters form a small group at the base of the stipule [14]. These variations can serve as taxonomic features at the species level.

Various types of colleters have been identified in Rubiaceae, characterized by their morphology [3,5,18,27]. The colleters in the studied species feature a palisade secretory epidermis and a parenchymatous central axis, anatomically aligning with the standard type, as Thomas [3] and Da Cunha and Vieira [28] described. This alignment supports findings in various species within the same family [7,14,27,29,30,31,32]. Furthermore, regarding morphology, the standard colleter outlined by Lersten [27] has a slightly wider medial portion than the apex and the base. The colleters described here are cylindrical, hence they are called cylindriform, a nomenclature previously adopted by Tullii et al. [6]. The central axis formed by parenchymatic cells in *P. tetraphylla* and *P. rudgeoides* suggests that these colleters have a mixed origin, deriving from the protoderm and the ground meristem. This confirms the definition of this structure as an emergence, a terminology discussed by Rio et al. [33], Klein et al. [14], and Paiva and Machado [34]. 

The colleters in this study exhibited vascularization and crystals in their central axis. The supported hypothesis is that the vascular bundle in the colleters represents a continuity of the vascularization of the stipule [3,27]. In Apocynaceae, the presence of this vascular bundle in the colleters is considered an evolutionary step in the family [29]. The vascularization of colleters in Rubiaceae has been described in various genera, such as *Bathysa* [5], *Cephalanthus* [19], *Simira* [14], *Mitracarpus,* and *Staelia* [31]. However, more research is needed to address this aspect and adequately interpret the vascularization of colleters in Rubiaceae.

The presence of crystals in the central axis of the studied species is in line with frequent reports from colleters from various Rubiaceae species [21,27,30,35,36,37]. In the Rubioideae subfamily, raphid crystals are indicated as a typical feature [3,27]; these were identified in both studied species. Additionally, styloid crystals were also observed in *P. tetraphylla*, occupying the same tissue, and this information is relevant for systematic groupings. A function often associated with crystals is the relationship of crystals with calcium metabolism and ionic balance, as well as the removal of toxic oxalate accumulations [38]. 

Some authors propose that secretion travels along the outer periclinal walls of the secretory cells of the colleters through secretion accumulation sites [7,8,39]. Protrusions of various sizes on the walls of *P. tetraphylla* colleters and prominent swellings on the surface of *P. rudgeoides* colleters, along with secretion accumulation in the intercellular spaces of both species, in the subcuticular space of *P. rudgeoides* and the cuticular layers of both species, suggest that the secretory process occurs through the wall, a result of the pressure exerted by secretion accumulation [40,41]. Paiva [42] emphasizes that the accumulation of secretion products between the cell wall and the cuticle generates the pressure that allows the secretion flow to pass through the cuticular barrier. Thus, secretion can be released without requiring energy. Evidence indicates that the colleter secretion mechanism may vary among species, as observed in *Cnidoscolus pubescens* Pohl., where the colleters, both in petiolar glands and in stipules, release their contents into the external environment by breaking the cuticle [43]. In addition to the various mechanisms for releasing secretion during colleter activity, it is believed that the premature senescence of these structures may be related to a mechanism for releasing accumulated secretion in the intercellular and periplasmic spaces [44].

The secretion of colleters can have a complex chemical composition, often characterized as mucilaginous [10,16,41,43,44]. The colleters of the analyzed species produce a secretion rich in acidic mucopolysaccharides, related to the protective function against desiccation in plant cells [6,11,34], reinforcing the role of colleters in the abiotic defense of these structures. Various researchers have also identified the presence of proteins in the colleters secretion [14,43,44,45]. Studies, such as those by Vieira et al. [46], Miguel et al. [4], and Tullii et al. [15] on Rubiaceae colleters highlight the importance of these molecules in plant defense, reinforcing the role of colleters in the biotic defense of these structures. 

Lipids were not detected in the studied species, contrasting with previous findings of lipids in the secretion of colleters in Rubiaceae species found in savannas [16]. Tresmondi et al. [16] noted that this corresponded to the chemical nature of their secretions, with forest species predominantly secreting hydrophilic substances and savanna species secreting lipophilic or mixed substances. This aligns with our findings and underscores the role of colleters in the abiotic responses of these structures.

Phenolic compounds were exclusively detected in the tissue of senescent colleters of *P. tetraphylla*. Numerous studies have noted the association between senescence and phenolic compound accumulation [5,41,42,43,47,48]. We suggest that during the colleters senescence, there is an increase in the concentration of antioxidants, such as phenolic substances, helping in the mitigation of oxidative stress linked to cellular aging. This underscores another role of colleters in the abiotic responses of these structures.

The senescence of colleters in the examined species was also accompanied by changes in anatomical, ultrastructural, and biochemical features, with these variations distinct among the studied species. Similar to what was observed in *P. tetraphylla*, other species also exhibit colleters at different stages of development in the same stipule [12,14,22,43]. The occurrence of colleters at different stages of development may be attributed to the asynchronous development of colleters or may represent a strategy to ensure secretion throughout the development of primordia [49].

In *P. rudgeoides*, the stipular nodes exhibited colleters that appeared to be in the same stage, indicating that younger stipules exclusively had colleters in the early stage, while senescence progressed gradually along the nodes, initially evidenced by a change in coloration. This sequence of color changes associated with colleters’ senescence was similarly observed in other species [3,6,29,50,51,52], possibly linked to ultrastructural alterations and the presence of phenolic compounds during the senescence phase.

Secretory cells typically display dense cytoplasm characterized by an abundance of mitochondria, small vacuoles, and a Golgi complex, indicating high metabolic activity. The frequency of other organelles varies depending on the secreted material [7,53]. In *P. tetraphylla*, young colleters exhibited turgidity and an intact appearance in the epidermis and parenchyma. However, in the ultrastructure, it was observed that the same colleter had cells with entire cytoplasm and others with utterly degraded cytoplasm due to the presence of bacteria in some cells, likely causing cytoplasmic degradation. Apart from these cells, the features observed in young colleters included peripheral cytoplasm with a large vacuole, a conspicuous nucleus with an intact envelope, rough endoplasmic reticulum, mitochondria, and numerous plastids. Regarding the secretory cells of young colleters in *P. rudgeoides*, the observed differences included the presence of a less developed Golgi complex, plasmodesmata, lipid droplets in the cytoplasm, and numerous mitochondria clustered at the nuclear periphery. 

The presence of peripheral cytoplasm with a large vacuole in the cells of the studied colleters may result from the fusion of small vacuoles that existed in earlier stages of development. Additionally, the observation of a poorly developed Golgi complex in the secretory cells of *P. rudgeoides* contradicts the frequent reports of hypertrophy of this organelle in colleters. The absence of the same organelle in the secretory cells of *P. tetraphylla* leads us to believe that it is less abundant in the colleters of this species, which is intriguing, as the abundant presence of the Golgi complex in the secretory phase of colleters has been described by various authors [6,14,40]. Studies also emphasize the importance of this organelle in the production of mucilage and proteins [54], compounds found in the secretion of the analyzed colleters. 

The notable characteristic of the cytoplasm in cells of *P. rudgeoides* was the clustering of mitochondria and rough endoplasmic reticulum near the nucleus. Studies on the ultrastructure of colleters emphasize the significant presence of endoplasmic reticulum, predominantly in the perinuclear region [54]. This localization is intrinsically linked to the role of the reticulum in protein synthesis, and the concentration of mitochondria in the same region is considered justified due to the high energy demand required for protein production. In the case of the analyzed colleters, these proteins constitute a part of the secreted product. 

Despite ultrastructural observations, it is essential to emphasize that knowledge about programmed cell death processes in plants is relatively limited, and morphological evidence may not be sufficient for a comprehensive discussion of cell death events [44]. However, through the TUNEL assay, more detailed information about this process was obtained. In the first node, fragmented nuclear DNA was observed; in subsequent nodes, it had already undergone degradation. These characteristics indicate that the secretory cells of *P. rudgeoides* colleters undergo non-autolytic programmed cell death, as the cells die prematurely, even before the onset of cytoplasmic degradation. Tullii et al. [6] characterized the same type of programmed cell death in colleters of *Alseis pickelii* Pilg. & Schmale. Nuclear DNA fragmentation was detected from the first node of the stipules, with a gradual decrease in TUNEL assay labeling in subsequent nodes. Ultrastructural features included cytoplasmic retraction, which was not peripheral as observed in the studied species, followed by organelle degradation.

Microorganisms are frequently identified in the secretion of colleters, especially in plants inhabiting environments as rich in the microbiota as the Atlantic Forest [15]. In Rubiaceae, it is proposed that colleters’ secretion plays a role in the nutritional aspects of bacterial nodular leaf symbiosis [27], exemplifying a type of relationship known as mutualism, in which plants secrete nutrients to specific groups of animals, gaining benefits in return. In the specimens analyzed, microorganisms were detected even within the secretory cells, including young colleters in the species *P. tetraphylla*. The results reveal that the dynamics of colleters can create a favorable environment for the proliferation of beneficial organisms [15]. Considering the composition of the secretion in these species, the absence of repellent compounds is observed, reinforcing the suggestion that the polysaccharides present in the secretion play a role in nourishing other organisms, in addition to regulating the hydration of these vegetative structures [43].

## 4. Materials and Methods

### 4.1. Plant Material

The sample collection was conducted in the Itatiaia National Park (Itatiaia, RJ, Brazil), a protected area located in the states of Rio de Janeiro and Minas Gerais, in May 2014. Established in 1937, it is Brazil’s first national park, encompassing an area of approximately 30 thousand hectares. The region is the Atlantic rainforest, renowned for its ecological significance, hosting endemic species [55].

The collected plant material comprised the apical shoots apex and stipules of *P. rudgeoides* and *P. tetraphylla*. Due to the deciduous nature of the stipules, it was possible to collect three nodes of stipules for *P. tetraphylla* and five nodes of stipules for *P. rudgeoides*. Four individuals of *P. rudgeoides* and five individuals of *P. tetraphylla* were collected.

### 4.2. Sample Preparation for Different Microscopy Techniques

Apical shoots and stipules were chemically fixed in aqueous solution containing 2.5% glutaraldehyde, 4.0% formaldehyde, and 0.05 M sodium cacodylate buffer at pH 7.2 at room temperature, at the collection site. Subsequently, fragments of stipules containing colleters were washed three times for 45 min in the same buffer and post-fixed for one hour in a solution of 1% osmium tetroxide and 0.1 M sodium cacodylate buffer, 0.05 M at pH 7.2, at room temperature. Subsequently, samples were washed in the same buffer and dehydrated in a series of acetone. Later, some fragments were infiltrated with epoxy resin (Embed 812), using an increasing series of resin in acetone. Resin polymerization was performed at 60 °C, for 48 h.

For scanning electron microscopy (SEM), after dehydration, samples were critical point dried (CPD 030 Bal-Tec^®^, Los Angeles, CA, USA). Then, the fragments were mounted on sample holders using double-sided adhesive tape and covered with a thin layer of 20 nm gold (SCD 050 Bal-Tec^®^, CA USA). Samples were examined under a scanning electron microscope (DSM 962 Zeiss^®^,Oberkochen, Germany) at 25 kV under standard secondary detection.

For light microscopy (LM), after polymerization of the resin, semi-thin sections were made in the material with thicknesses of 0.60 and 0.70 μm) in an ultramicrotome (Reichert Ultracuts Leica Instruments^®^, Wetzlar, Germany) with a diamond blade (Diatome^®^, Nidau, Switzerland). The sections were placed on slides, stained with 0.05% toluidine blue, and mounted with coverslips and Entellan^®^ (Merk Millipore, MA, USA). Serial sections were examined and images were obtained with a Zeiss^®^ Axioplan equipped with a 14-megapixel Cannon Power Shot digital camera.

Alternatively, fixed stipules were dehydrated in an ethanol series, infiltrated, and embedded in Historesin (Leica Instruments^®^, Wetzlar, Germany) for histochemical tests. Successively, transverse and longitudinal 3.0 μm sections of the material were obtained using the CUT 4050 microtome (SLEE Technik GmbH, Mainz, Germany). The sectioned material was placed on slides, and the reagent used to identify pectin compounds and mucopolysaccharides was an aqueous solution of 0.02% Ruthenium Red [56]. For the detection of acidic and basic compounds, an aqueous solution of 0.05% toluidine blue was used [57]. The cuticle was revealed with an aqueous solution of 0.01% Auramine O, followed by observation under fluorescence microscopy (filter excitation from 470 to 490 nm and emission from 515 to 565 nm) (Heslop-Harrison and Knox 1981, adapted). Lipids were detected by 0.03% Sudan IV [58], while the NADI reagent was used to detect essential oil [59]. Proteins were treated using 0.25% Coomassie Brilliant Blue [60]. A 10% ferric III chloride reagent was applied to identify phenolic compounds [61]. To reveal fatty acids, sections were treated sequentially with 0.05% copper acetate for 3 h and 0.1% EDTA in 0.1 M sodium phosphate buffer, pH 7.1, for 5 min, and 0.1% rubeanic acid in 70% ethanol for 30 min [62]. The tests were conducted in triplicate, and the semi-permanent slides were mounted with glycerin and a coverslip. They were observed under fluorescence microscopy Zeiss^®^ Axioplan equipped with a 14-megapixel Cannon Power Shot digital camera.

For transmission electron microscopy (TEM), ultrathin sections of approximately 70 nm thick were obtained in an ultramicrotome (Reichert Ultracuts Leica Instruments^®^, Wetzlar, Germany) with a diamond blade (Diatome^®^, Nidau, Switzerland). The cuts were collected on 300 mesh copper grids. Sections were stained with 5% uranyl acetate for 40 min and lead citrate for 5 min at room temperature and then observed under the transmission electron microscope (TEM 900 Zeiss^®^, Oberkochen, Germany) at 80 KV.

### 4.3. DNA Fragmentation Analysis

The stipules were cut into small fragments and fixed in 50% FAA for two days. Samples were washed in 50% ethanol, dehydrated in an ascending series of ethanol, and embedded in Historesin (Historesin Leica Instruments, Wetzlar, Germany). Subsequently, 3.0 μm transverse and longitudinal sections were obtained using the CUT 4050 microtome (SLEE Technik GmbH, Mainz, Germany). According to the manufacturer’s instructions, the TUNEL assay was performed using the in situ cell death detection kit (TMR Red, Roche Applied Science, Penzberg, Germany). The sections were incubated with proteinase K (SIGMA, St. Louis, MS, USA) for 20 min. Subsequently, the TUNEL kit reaction mixture was added, and the sections were incubated at 37 °C for 60 min. The sections were washed in PBS, and the samples were analyzed by fluorescence microscopy at excitation wavelengths of 520 to 560 nm and emission wavelengths of 570 to 620 nm, thus detecting free 3’-OH groups at the fragmented ends of DNA. A negative control was performed without adding the solution with terminal deoxynucleotidyl transferase. The assays were performed in triplicate.

## 5. Conclusions

The analysis of colleters in *P. tetraphylla* and *P. rudgeoides* reveals the complexity of these structures and their functional importance in various contexts. Through anatomical, histochemical, senescence, and ultrastructural analyses, the study unveils colleters characterized by a palisade epidermis and parenchymatic central axis, classified as the standard type, exhibiting vascularization and crystals. The secretion of colleters, rich in acid mucopolysaccharides, proteins, and phenolic compounds, not only shields against desiccation but also participates in the plant biotic responses. The ontogenesis, development, and senescence of colleters are quite rapid; however, we observed colleters present at different stages of development in the different nodes, ensuring biotic and abiotic protection to the shoot apex and axillary buds. Pronounced bulges on the surface of colleters, coupled with the accumulation of secretion in intercellular and subcuticular spaces, suggest a secretory process occurs through the cell wall, driven by pressure from the secretion accumulation. The presence of microorganisms in the colleters secretion, especially in microbiota-rich environments like the Atlantic Forest, provides valuable insights into plant–microorganism interactions. The morphoanatomical characteristics of colleters, the complexity of mixed secretion, the peculiarities in the senescence process, and the secretion mechanism highlight the adaptation of these plants to specific environments and their intricate interactions with the environment and other organisms. 

## Figures and Tables

**Figure 1 plants-13-01206-f001:**
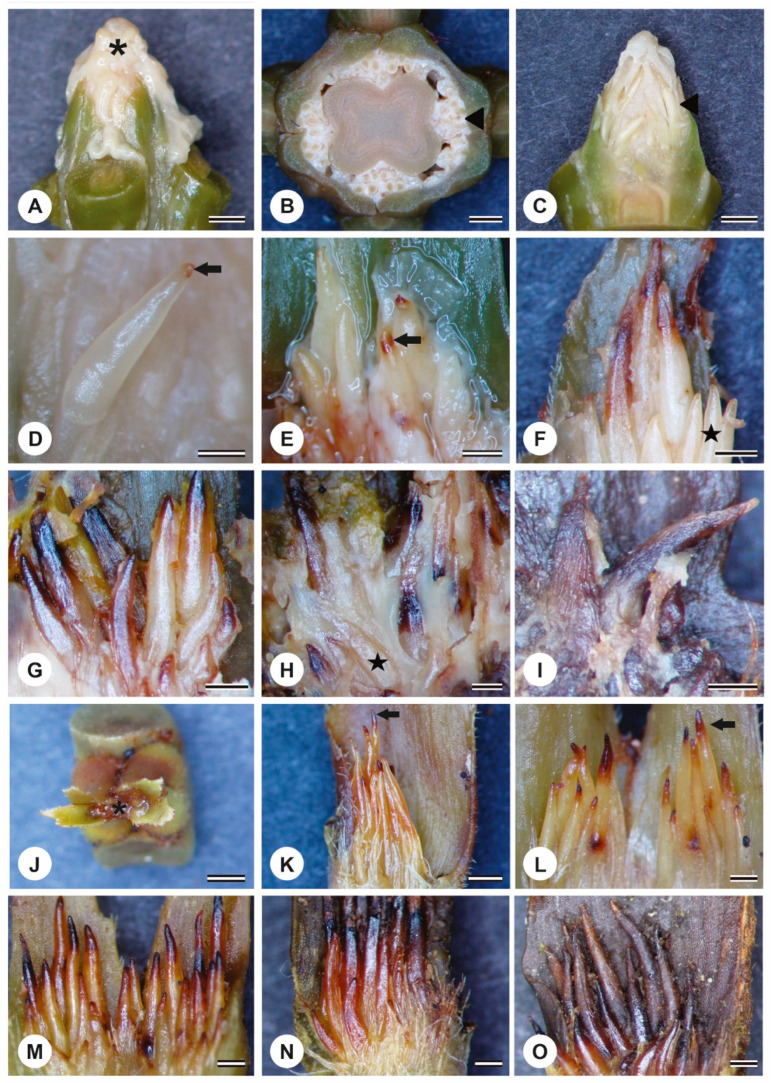
Stereomicroscopy of the shoot apex and colleters of *P. tetraphylla* (**A**–**I**) and *P. rudgeoides* (**J**–**O**). (**A**) shoot apex covered with secretion (*). (**B**) transverse section of the shoot apex. Note the colleters (triangle). (**C**) colleters of the first node. Note the white color (triangle). (**D**) colleters of the first node show senescence at the apex (arrow). (**E**,**F**) colleters in the second node showing advancement of senescence from the apex to the base (arrow) and young colleters (star). (**G**–**I**) colleters in the third node with dark color. Note advanced senescence and others with white color (star). (**J**) secretion at the apex of the shoot (*). (**K**,**L**) colleters of the first and second nodes, respectively. Note the senescence at the apex (arrow). (**M**–**O**) vests, third (**M**), fourth (**N**), and fifth (**O**). Note the advancement of senescence from the apex to the base. Scale bars: (**A**–**C**) and (**J**), 1 mm; (**D**), 0.25 mm; (**E**,**I**) and (**K**–**O**), 0.5 mm.

**Figure 2 plants-13-01206-f002:**
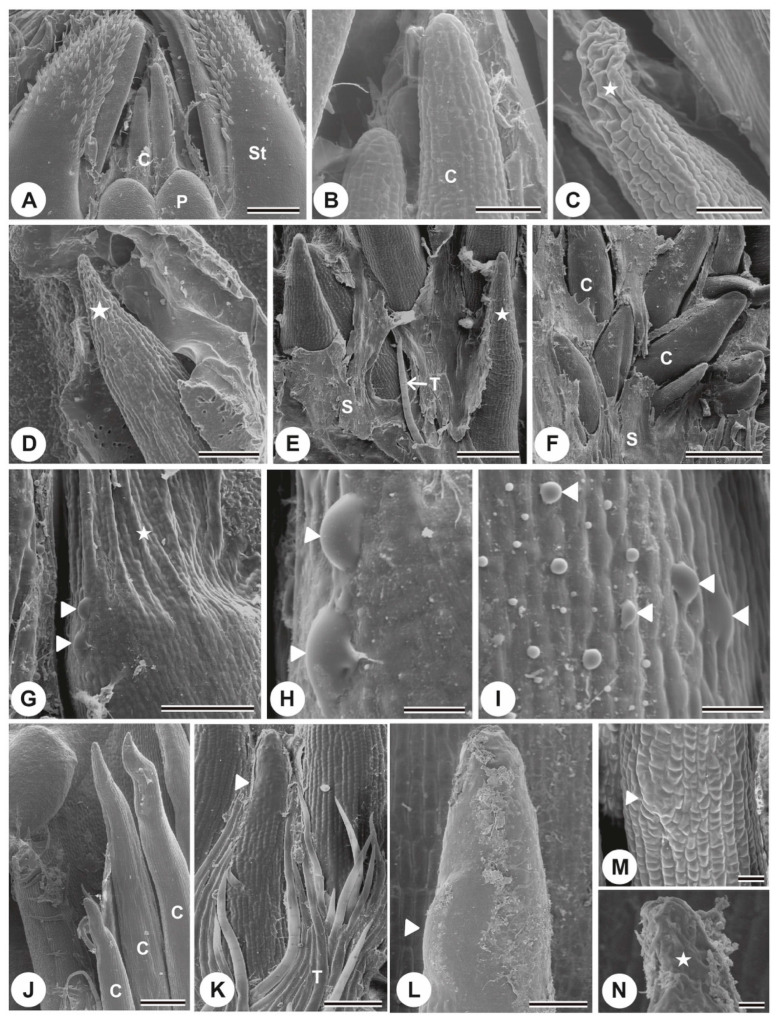
SEM of the stem apex and colleters of *P. tetraphylla* (**A**–**I**) and *P. rudgeoides* (**J**–**N**). (**A**) colleters from the first node emerge from the base of the stipules. (**B**) detail of the colleters, highlighting their turgid surface. (**C**,**D**) colleters from the second and third nodes, respectively, with wrinkled apices (star). (**E**) colleter from the third node completely wrinkled (star). (**F**) turgid colleters from the third node. (**G**–**I**) colleters with prominences on their cell walls (arrowhead). (**J**) frontal view of the colleters from the first node. Note the cell contours through the cuticle. (**K**–**M**) colleters from the second, third, and fourth nodes displaying cuticle distension caused by secretion accumulation in the subcuticular space (arrowhead). (**N**) colleter from the fifth node with a wrinkled apex (star). C = colleter, S = secretion, St = stipule, P = leaf primordium, T = protective trichomes. Scale bars: (**A**,**C**) 200 μm; (**B**,**E**,**M**) 50 μm; (**D**,**G**,**L**) 100 μm; (**F**) 500 μm; (**H**,**I**,**N**) 20 μm; (**J**) 250 μm; (**K**) 25 μm.

**Figure 3 plants-13-01206-f003:**
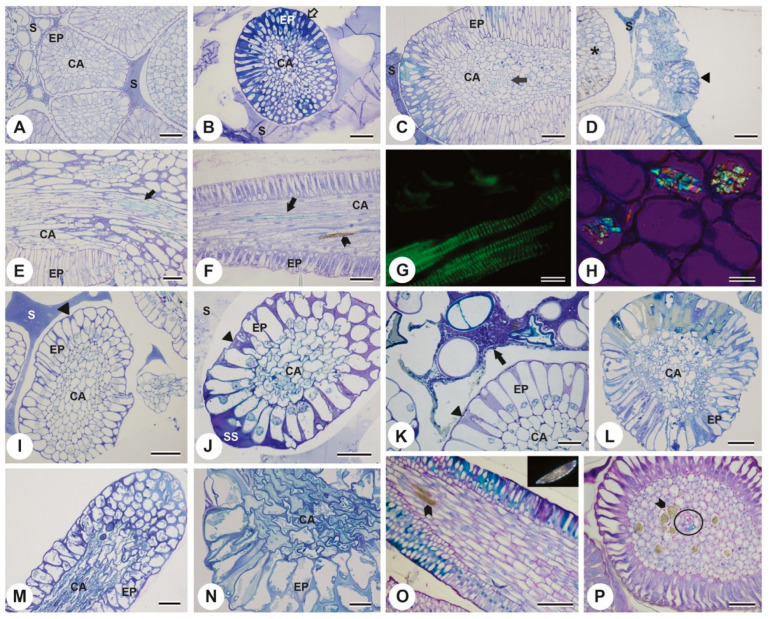
Anatomy of the colleters of *P. tetraphylla* (**A**–**H**) and *P. rudgeoides* (**I**–**P**) revels with toluidine blue. (**A**) cross-section of a colleter from the first node highlighting a central axis surrounded by palisade epidermis and secretion accumulation. (**B**) cross-section of a colleter from the second node showing secretion accumulation in the intercellular spaces (empty arrow). (**C**) colleter from the second node exhibiting vascularization in the central axis (CA). (**D**) cross-section of colleter from the second node, showing the epidermis of a young colleter (asterisk) and a senescent colleter (triangle). (**E**) longitudinal section of a colleter from the third node highlighting vascularization at its base (full arrow). (**F**) colleter with vessel element in the central axis (arrow) and raphid crystals (arrowhead). (**G**) detail of vessel elements in the central axis of a colleter in fluorescence microscopy. (**H**) styloid crystals under polarized light in the central axis of a colleter. (**I**) cross-section of a colleter from the first node highlighting a central axis surrounded by palisade epidermis and secretion accumulation both on the exterior and in the intercellular space (arrowhead). (**J**,**K**) cross-sections of colleter from the second node highlight increased secretion accumulation in the intercellular spaces (arrowhead) and the subcuticular space (SS). (**K**) note microorganisms in the secretion accumulating around the colleter (arrow). (**L**–**N**) cross-sections of colleter from the third, fourth, and fifth nodes, respectively, with disorganized and degraded epidermis and a central axis, highlighting the senescence process of the colleter. (**O**) longitudinal section highlighting crystals in the central axis of the colleter (arrowhead). Note a crystal under polarized light in the insert. (**P**) cross-section of a colleter with several crystals in its central axis (arrowhead); note the vascularization (circle). Ep = epidermis, CA = central axis, S = secretion, SS = subcuticular space. Scale bars: (**A**,**B**,**D**,**E**) 40 μm; (**C**–**F**,**H**,**K**,**M**,**N**,**O**,**P**) 20 μm; (**G**) 10 μm. (**I**,**J**,**L**) 50 μm.

**Figure 4 plants-13-01206-f004:**
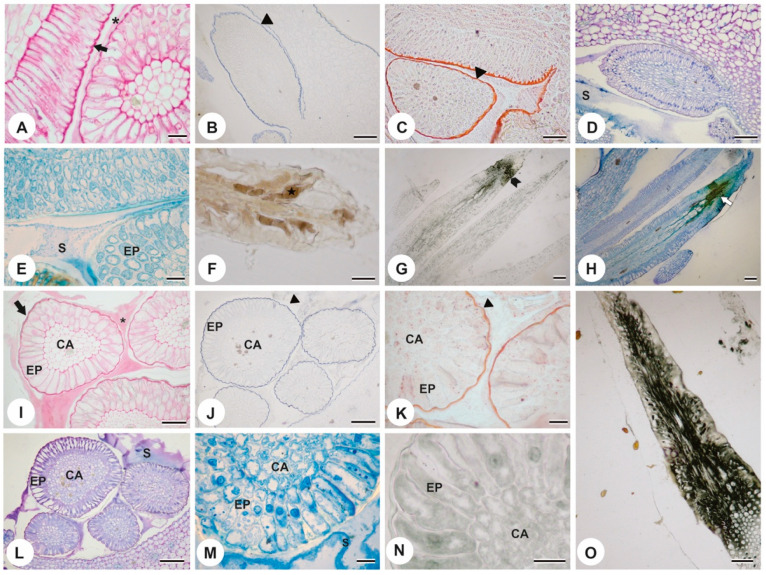
Histochemical tests of the colleters of *P. tetraphylla* (**A**–**H**) and *P. rudgeoides* (**I**–**O**). (**A**) Ruthenium Red test on a young colleter showing a rich polysaccharide layer on the wall (arrow) and labeling in the secretion (asterisk). (**B**,**C**) tests with Nadi reagent (**B**) and Sudan IV (**C**) showing that the colleter is covered by cuticle (triangle). (**D**) young colleter stained with toluidine blue, highlighting the presence of basic substances in the secretion. (**E**) Coomassie Brilliant Blue test showing the presence of proteins. (**F**–**H**) tests showing the accumulation of phenolic compounds (**F**), fatty acids (**G**), and acidic substances (**H**) (white arrow) in senescent colleters. (**I**) Ruthenium Red test on a young colleter showing a rich polysaccharide layer on the wall (arrow) and labeling in the secretion (asterisk). (**J**,**K**) tests with Nadi reagent (**J**) and Sudan IV (**K**) showing that the colleter is covered by cuticle (triangle). (**L**) young colleter stained with toluidine blue, highlighting the presence of basic substances in the secretion. (**M**) Coomassie Brilliant Blue test showing the presence of proteins. (**N**) negative test for fatty acids in young colleters. (**O**) positive test for fatty acids in senescent colleters. Ep = epidermis, CA = central axis, S = secretion. Scale bars: (**A**,**J**,**L**) 50 μm; (**B**,**D**,**H**,**O**) 100 μm; (**C**,**E**,**F**,**I**,**M**,**N**) 20 μm; (**G**) 200 μm.

**Figure 5 plants-13-01206-f005:**
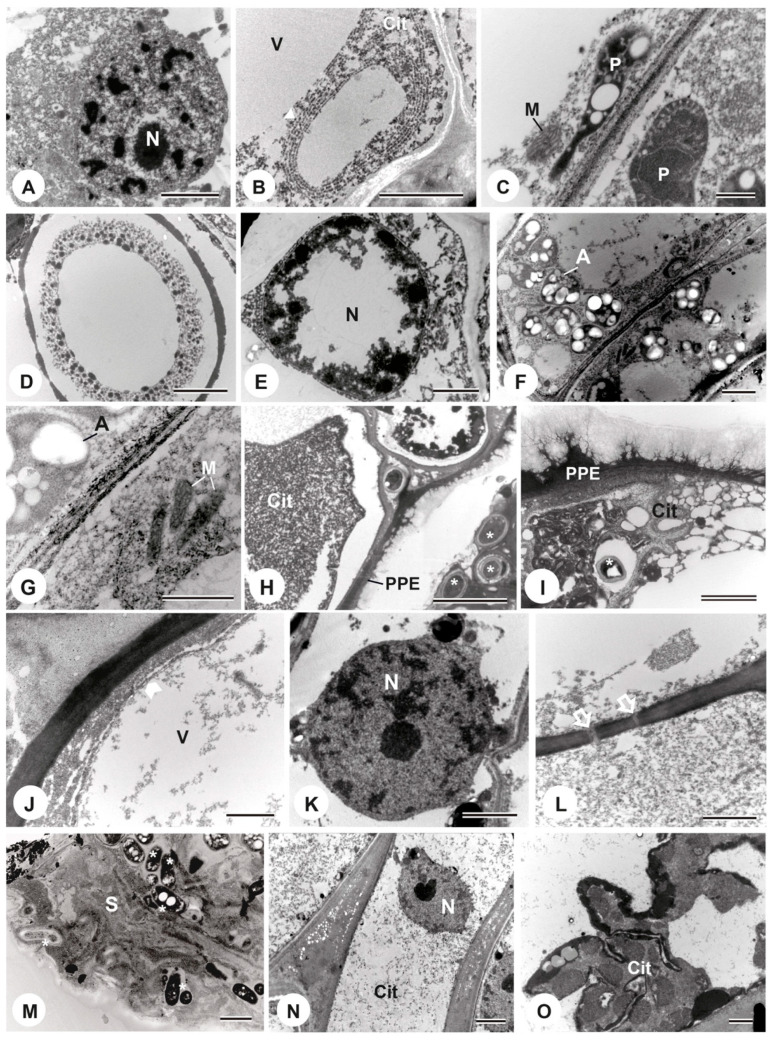
TEM images of the ultrastructure of secretory cells colleters of *P. tetraphylla* (**A**–**I**) and *P. rudgeoides* (**J**–**O**), highlighting the senescence process. (**A**–**C**) secretory cells of colleters from the first node with conspicuous nucleus (**A**), large vacuole, and rough endoplasmic reticulum (arrowhead) (**B**), plastids, mitochondria (**C**), and microorganism (**D**). (**E**) secretory cell from the second node showing a nucleus with irregularly distributed condensed chromatin and structurally altered envelope. (**F**) secretory cell from the third node with the accumulation of amyloplasts and several mitochondria. (**G**) detail of micrograph (**F**). (**H**,**I**) secretory cells from the third nodes, with microorganisms (*) inside and completely degraded cytoplasm. (**J**–**L**) secretory cells from the first node with peripheral cytoplasm and vacuole with intact membrane (arrowhead) (**J**), conspicuous nucleus (**K**), plasmodesma (empty arrow) (**L**). (**M**) secretion with numerous microorganisms (*). (**N**) secretory cell from the fourth node with degraded cytoplasm and a deformed nucleus with a ruptured envelope. (**O**) secretory cells from the fifth nodes, with completely degraded cytoplasm. A = amyloplast, Cit = cytoplasm, M = mitochondrion, N = nucleus, P = plastid, PPE = outer periclinal wall, S = secretion. Scale bars: (**A**,**B**,**E**,**F**,**I**,**K**) 2 µm; (**C**) 500 nm; (**D**,**G**,**J**,**L**,**M**,**O**) 1 µm; (**H**) 5 µm; (**N**) 2.5 µm.

**Figure 6 plants-13-01206-f006:**
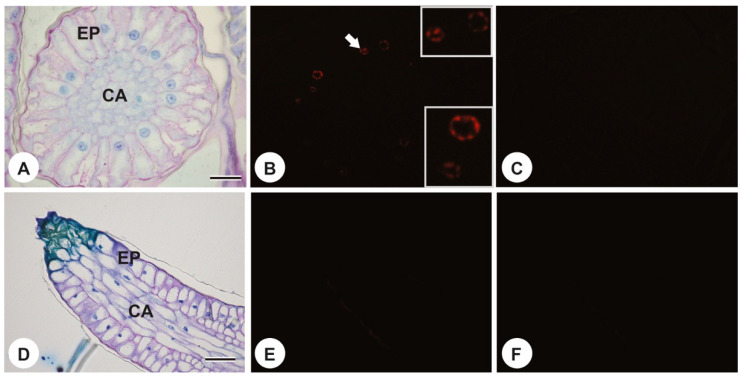
TUNEL assay on *P. rudgeoides* colleters. (**A**–**C**) first-node colleters in bright field, TUNEL assay, and negative control, respectively. (**B**) TUNEL-positive showed that they already had highly fragmented DNA (white arrow). (**D**–**F**) fourth-node colleters in bright field, TUNEL assay, and negative control, respectively, showing TUNEL-negative nuclei in (**E**). EP = Epidermis, CA = central axis. Bars: (**A**–**C**)—20 µm, (**D**–**F**)—50 µm.

## Data Availability

All data supporting the findings of this study are available within the paper; if any query remains, please contact the author for correspondence.

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
