# Peer review of "Unveiling the Anatomical and Functional Attributes of Stipular Colleters in Palicourea tetraphylla Cham. & Schltdl. and Palicourea rudgeoides (Müll. Arg.) Standl. (Rubiaceae)"

_plants, 2024, doi:10.3390/plants13091206_

Round 1

Reviewer 1 Report

Comments and Suggestions for Authors

Please see the comments in the attachment

Comments on the Quality of English Language

Minor editing of English language is required

Author Response

Thank you very much for taking the time to review this manuscript. Please find the detailed responses and the corresponding corrections highlighted in the re-submitted files.

Reviewer 2 Report

Comments and Suggestions for Authors

The paper reports a histochemical and ultrastructural study of colleters, a type of multicellular secretory structures, in two members of the Rubiaceae family. Despite a series of bold statements in the abstract, the paper provides virtually no novel information on the topic, which has been extensively investigated in the past by the same researchers as well as others. Good transmission electron microscopy might have added an extra value, but unfortunately the figures included in the paper reflect fixation problems and are uninformative.

Comments on the Quality of English Language

The quality of English was acceptable, but the manuscript requires some editing (repetitions and badly structured sentences). This, provided the authors add extra scientific relevance.

Author Response

Thank you very much for taking the time to review this manuscript. We have reviewed the entire manuscript and made several improvements to the text and images. Additionally, the English in the whole manuscript has been thoroughly reviewed.
This study complements the work previously conducted by our group. Despite being a widely studied topic, the description of the secretion mechanism of the colleter in the species studied allowed for an extensive discussion on how this process has been viewed and interpreted by different authors.

Reviewer 3 Report

Comments and Suggestions for Authors

In the manuscript entitled: "Structure and senescence of stipular colleters in Palicourea tetraphylla Cham. & Schltdl. and Palicourea rudgeoides (Müll. Arg.) Standl. (Rubiaceae)" (ID number: plants-2872747), the authors presented the anatomical and ultrastructural structure of stipular colleters, including the content of secretions stored in them, in two species of the Rubiaceae family.

I found a few of problematic points in the manuscript, some of which I quote below.

Introduction:

- “…and adaptability to plants in various environmental conditions.” – it should be “of plants”.

- “…a wide diversity of alkaloids in the genus Palicourea …” – incomprehensible sentence, where are these alkaloids located?

Results:

- “…However, most secretory cells had microorganisms inside …” – please explain what “microorganisms” the authors observed.

Materials and Methods:

Plant material:

- No information on the abundance of the population studied (how many individuals were in the population studied) and how many replicates of observations and analyses were made.

 In my opinion, the topic of the work is important from the point of view of developmental biology and plant taxonomy. It should be emphasized that the authors presented very good documentation from many types of microscopes, which definitely makes the results more credible.

Author Response

(The authors gave the same response as above.)

Round 2

Reviewer 2 Report

Comments and Suggestions for Authors

I found no significant improvement in the scientific relevance of the paper, nor in the quality of presentation. I suggest the authors to improve the ultrastructural analysis. Use freshly-made aldehyde fixative from sealed ampoules, and adjust the pH of the final solution to 7.4 just before use. Add 0.5% potassium ferricyanide to the osmium fix to enhance membrane preservation. The Tunel test is clearly faulty: I hardly believe that the nuclei visible in figs 7D,G,J do not contain DNA, and by the way even the test made on young colleters shows almost no reaction. The hypothesis that the secrete passes across the cell wall, emphatically reported in the Abstract, is trivial. I could see no other way for the secrete to exit the cells, unless the cell walls were broken, which is not the way plant glands work. Indeed, fig. 5 I illustrates the process quite well.

Comments on the Quality of English Language

The manuscript retains grammar errors and akward sentences

Author Response

Dear reviewer, thank you for your comments. We have diligently sought to identify and implement the suggested improvements. We have changed the approach to the TEM results, focusing on the senescence process and the evidence of microorganisms, which is the novelty discussed in the discussion section.

Regarding the TUNEL test, we present below two previously published works using the same technique, demonstrating that this experiment is not a fraud. In fact, these two articles, with co-authors even from the international scene, include DNA quantification. In this figure, we have included only the first node and the completely senescent node. Below is the citation of the works.

1)      Tullii, CF; Miguel, EC; Lima, NB; Fernandes, KVS; Gomes, VM; Da Cunha, M. Characterization of stipular colleters of Alseis pickelii. Botany, v. 91, p. 403-413, 2013.

2)      Lima NB; Trindade FG; Da CUNHA, M; OLIVEIRA, AEA; Topping, J; Lindsey, K; Fernandes, KVS. Programmed cell death during development of cowpea (Vigna unguiculata) (L.) Walp.) seed coat. Plant, Cell and Environment (Print), v. 38, p. 718-728, 2015

Any further questions, I am available.

Round 3

Reviewer 2 Report

Comments and Suggestions for Authors

The novel version is not significantly better that the former. I confirm my opinion that the paper does not provide any useful information on the topic.

Comments on the Quality of English Language

The paper retain several grammar errors

Author Response

We have endeavored to implement all suggested improvements within our means. We are aware of the significance of our research and how these findings will be relevant to the field.